# Fast and Effective Robustness Certification

**Gagandeep Singh, Timon Gehr, Matthew Mirman, Markus Püschel, Martin Vechev**
Department of Computer Science
ETH Zurich, Switzerland
`{gsingh,timon.gehr,matthew.mirman,pueschel,martin.vechev}@inf.ethz.ch`

## Abstract

We present a new method and system, called *DeepZ*, for certifying neural network robustness based on abstract interpretation. Compared to state-of-the-art automated verifiers for neural networks, *DeepZ*: (i) handles ReLU, Tanh and Sigmoid activation functions, (ii) supports feedforward, convolutional, and residual architectures, (iii) is significantly more scalable and precise, and (iv) and is sound with respect to floating point arithmetic. These benefits are due to carefully designed approximations tailored to the setting of neural networks. As an example, *DeepZ* achieves a verification accuracy of $97\%$ on a large network with $88,500$ hidden units under $L_\infty$ attack with $\epsilon = 0.1$ with an average runtime of $133$ seconds.

## 1   Introduction

Neural networks have become an integral part of many critical applications such as vehicle control, pattern recognition, and medical diagnosis. However, it has been shown recently that neural networks are susceptible to *adversarial attacks*, where the network can be easily tricked into making wrong classification by only slightly modifying its inputs [26, 12, 17, 3, 2, 9, 23, 22]. As a result, there is considerable interest in formally ensuring robustness of neural networks against such attacks.

Robustness verifiers can be complete or incomplete. Complete verifiers do not have false positives but have limited scalability as they are based on computationally expensive methods such as SMT solving [14, 8], mixed integer linear programming [27] or input refinement [28]. On the other hand, incomplete verifiers can produce false positives but they scale better than complete verifiers. Incomplete verifiers employ a variety of methods including duality [24, 7], abstract interpretation [10], and linear approximations [29, 15].

Orthogonal to robustness certification, *adversarial training* focuses on making neural networks robust by training against an adversarial model of attack. Gu and Rigazio [13] add concrete noise to the training set and remove it statistically for defending against adversarial examples. Goodfellow et al. [12] generate adversarial examples misclassified by neural networks and then design a defense against this attack by explicit training against perturbations generated by the attack. Madry et al. [19] show that training against an optimal attack also guards against non-optimal attacks. While this technique was effective in experiments, Carlini et al. [4] demonstrated an attack for the safety-critical problem of ground-truthing, where this defense occasionally exacerbated the problem. The work of [30] and [21] proposes scalable defenses against the standard $L_\infty$-based adversarial attacks.

In this paper we build on the work of Gehr et al. [10], which introduces the idea of using the classic framework of abstract interpretation [5] to soundly approximate the behavior of the network. Similar to [10], we also use the Zonotope abstraction for sound approximations. The Zonotope domain provides a closed affine form for each variable and enables a precise and cheap way to capture the effect of affine transformations inside neural networks, without requiring backpropagation as in [15, 29]. The key limitation of [10] however, is that it only provides a fairly generic abstract transformer for the ReLU activation function, which is slow and imprecise. Further, the work does not

discuss approximations of other important functions (e.g., Sigmoid, Tanh). Indeed, defining sound, scalable and precise abstract transformers is the most difficult aspect of abstract-interpretation-based analyses. While generic transformers tend to be easier to reason about and ensure soundness of, they often lack the scalability and precision of transformers that exploit the underlying properties of the abstract domain (e.g., Zonotope) and the function being approximated (e.g., ReLU).

**Our contributions.** In this work we address these limitations and make the following contributions:

- We introduce new, point-wise Zonotope abstract transformers specifically designed for the ReLU, Sigmoid, and Tanh activations often used by neural networks. Our transformers minimize the area of the projection of the zonotope to the 2-D input-output plane. Further, our transformers are sound w.r.t. floating point arithmetic.

- We implemented both, sequential and parallel versions of our transformers in an end-to-end automated verification system called *DeepZ*.

- We evaluated *DeepZ* on the task of verifying local robustness against $L_\infty$-norm based attacks on large MNIST and CIFAR10 feed forward, convolutional, and residual networks. In our evaluation we considered both, undefended as well as defended versions of the same network (defended against $L_\infty$ attacks using state-of-the-art defenses).

- Our experimental results demonstrate that *DeepZ* is more precise and faster than prior work. *DeepZ* precisely verifies large networks with $> 88,000$ hidden units under $L_\infty$-norm based perturbations within few minutes, while being sound w.r.t to floating point arithmetic.

To our best knowledge, *DeepZ* is currently the most scalable system for certifying local robustness of neural networks while guaranteed soundness w.r.t to floating point operations (used by all neural networks). All of our code, datasets and results are publicly available at http://safeai.ethz.ch/.

## 2 Abstract Interpretation for Verifying Robustness of Neural Networks

Abstract Interpretation [5] is a classic method for sound and precise over-approximation of potentially unbounded or infinite set of program behaviors. The key idea behind this framework consists of defining so called abstract transformers for statements used by the program (e.g., affine arithmetic, ReLU functions, etc). These transformers approximate (i.e., they are sound) the behavior of the statement by defining its effect on an abstract domain. An abstract domain is simply a set of abstract elements (approximations) typically ordered in a lattice of some kind.

A key challenge when defining abstract transformers is striking a balance between scalability (how fast the transformer computes the approximation) and precision (how much precision it loses). Once transformers are defined, the analysis with abstract interpretation proceeds by executing them on the particular program (e.g., a neural network) and computing a final approximation (a fixed point). The relevant property can then be checked on this final approximation: if the property can be proved, then it holds for any concrete input to the program, otherwise, it may either hold but the abstraction was too coarse and unable to prove it (i.e., a false positive) or it may indeed not hold.

Verifying robustness properties of neural networks exactly is computationally expensive as it usually requires evaluating the network exhaustively on a prohibitively large set of inputs. Abstract interpretation can be leveraged for this problem by designing abstract transformers specifically for the computations used in the network, e.g., affine arithmetic and activation functions. The network can then be analyzed using these abstract transformers. For example, we can abstract a concrete input $\overline{x}$ and relevant perturbations to $\overline{x}$ (resulting in many different inputs) into one abstract element $\alpha_\mathcal{R}$ and then analyze the network starting from $\alpha_\mathcal{R}$, producing an abstract output $\alpha_\mathcal{R}^o$. We can then verify the robustness property of interest over $\alpha_\mathcal{R}^o$: if successful, it means we verified it over all concrete outputs corresponding to all perturbations of the concrete input.

In this paper, we consider local robustness properties $(\mathcal{R}_{\overline{x},\epsilon}, \mathcal{C}_L)$ where $\mathcal{R}_{\overline{x},\epsilon}$ represents the set of perturbed inputs around the original input $\overline{x} \in \mathbb{R}^m$ based on a small constant $\epsilon > 0$. $\mathcal{C}_L$ is a robustness condition which defines the set of outputs that all have the same label $L$:

$$C_L = \left\{ \overline{y} \in \mathbb{R}^n \ \mid \ \underset{i \in \{1,\ldots,n\}}{\arg\max}(y_i) = L \right\}.$$

A robustness property $(\mathcal{R}_{\overline{x},\epsilon}, \mathcal{C}_L)$ holds iff the set of outputs $\mathcal{O}_{\mathcal{R}}$ corresponding to all inputs in $\mathcal{R}_{\overline{x},\epsilon}$ is included in $\mathcal{C}_L$. $(\mathcal{R}_{\overline{x},\epsilon}, \mathcal{C}_L)$ can be verified using abstract interpretation by checking if the abstract output $\alpha_{\mathcal{R}}^o$ resulting from analyzing the network with an abstraction of $\mathcal{R}_{\overline{x},\epsilon}$ is included in $\mathcal{C}_L$.

**Zonotope Abstraction.** In this work, we build on the classic Zonotope numerical abstract domain, which we discuss below. This domain was already shown to be a suitable basis for analyzing neural networks by Gehr et al. [10]. In the next section, we introduce our new abstract transformers which leverage properties of the domain and are the novel contribution of this work.

Let $\mathcal{X}$ be the set of $n$ variables. The Zonotope abstraction [11] builds on affine arithmetic by associating an affine expression $\hat{x}$ for each variable $x \in \mathcal{X}$:

$$\hat{x} := \alpha_0 + \sum_{i=1}^{p} \alpha_i \cdot \epsilon_i, \quad \text{where } \alpha_0, \alpha_i \in \mathbb{R}, \epsilon_i \in [a_i, b_i] \subseteq [-1, 1] \tag{1}$$

This expression consists of a center coefficient $\alpha_0$, a set of noise symbols $\epsilon_i$, and coefficients $\alpha_i$ representing partial deviations around the center. Crucially, the noise symbols $\epsilon_i$ can be shared between affine forms for different variables which creates implicit dependencies and constraints between the affine forms. This makes the Zonotope abstraction more powerful than an Interval abstraction which only maintains ranges of a variable $x$. A range $[l_x, u_x]$ can be simply derived from the affine form by computing the minimal and maximal value possible.

A zonotope $\mathcal{Z} \subseteq \mathbb{R}^n$ is represented by a collection of affine forms for all variables $x \in \mathcal{X}$, and is the set of all possible (joint) values of the affine forms for an arbitrary instantiation of the noise symbols $\epsilon_i$. As in practice, it is impossible to compute with arbitrary real numbers, we instead use a slightly modified definition:

$$\hat{x} := [\alpha_0, \beta_0] + \sum_{i=1}^{p} [\alpha_i, \beta_i] \cdot \epsilon_i, \quad \text{where } \alpha_0, \beta_0, \alpha_i, \beta_i \in \mathbb{R}, \epsilon_i \in [a_i, b_i] \subseteq [-1, 1] \tag{2}$$

In this *interval affine form*, we have replaced all coefficients by intervals. All computations on intervals are performed using standard interval arithmetic. To ensure soundness with respect to different rounding modes and to account for the lack algebraic properties such as associativity and distributivity in the floating point world, the lower bounds and the upper bounds are rounded towards $-\infty$ and $+\infty$ respectively and suitable error values are added as explained in [20].

Since affine arithmetic is fast and exact for affine transformations, it is an attractive candidate for the verification of neural networks [10]. However, the Zonotope abstraction is inherently not exact for non-linear activation functions such as ReLU, Sigmoid, and Tanh. Thus, approximation is needed, which creates a tradeoff between the cost of computation and precision. As mentioned earlier, a generic approximation of the ReLU function was proposed by Gehr et al. [10], however, this approximation is both imprecise and costly as it relies on the expensive Zonotope join operator. Overall, this results in suboptimal precision and performance of the analysis.

## 3 Fast Zonotope Abstract Transformers

We now introduce our fast and precise *pointwise* Zonotope abstract transformers for the ReLU, Sigmoid, and Tanh activations (Sigmoid and Tanh are not supported by Gehr et al. [10]) and show their optimality in terms of area in the input-output plane. Our evaluation in Section 4 shows that our proposed approximations strike a good balance between precision and performance.

### 3.1 ReLU

The effect of applying the ReLU function on an input zonotope $\mathcal{Z}$ can be represented with the assignment $y := \max(0, x)$ where $x, y \in \mathcal{X}$. If $x$ can only have positive ($l_x > 0$) or non-positive values ($u_x \leq 0$) in $\mathcal{Z}$, then $\hat{y} = \hat{x}$ or $\hat{y} = [0, 0]$ respectively. The affine forms for the remaining variables are not affected and the resulting zonotope is exact. When $x$ can have both positive and negative values, then the output cannot be exactly captured by the zonotope abstraction and thus approximations are required. We define such an approximation for this case. The approximation can also be applied pointwise per layer, namely, only altering the affine form $\hat{y}$ while keeping all other affine forms in $\mathcal{Z}$ unaltered.

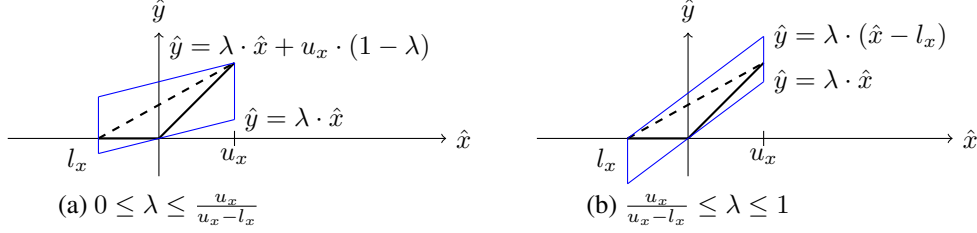

$$(a)\ 0 \leq \lambda \leq \frac{u_x}{u_x - l_x} \qquad\qquad (b)\ \frac{u_x}{u_x - l_x} \leq \lambda \leq 1$$

Figure 1: Two zonotope approximations for the ReLU function parameterized by the slope $\lambda$.

Fig. 1 shows the projections into the $xy$-plane of two sets of sound zonotope approximations. The projections have the shape of a parallelogram with two vertical lines and two parallel lines of slope $\lambda$, which is a parameter. To ensure soundness for all approximations in Fig. 1 (a), we require $0 \leq \lambda \leq \frac{u_x}{u_x - l_x}$. Similarly, $\frac{u_x}{u_x - l_x} \leq \lambda \leq 1$ for Fig. 1 (b). Notice that the two sets have one element in common at $\lambda = \frac{u_x}{u_x - l_x}$. Among the different candidate approximations in Fig. 1, we choose the one minimizing the area of the parallelogram in the $xy$-plane. The area $A_1(\lambda)$ of the parallelogram in Fig. 1 (a) is:

$$A_1(\lambda) = (1 - \lambda) \cdot u_x \cdot (u_x - l_x). \tag{3}$$

$A_1(\lambda)$ is a decreasing function of $\lambda$. Thus $A_1$ is minimized at $\lambda = \frac{u_x}{u_x - l_x}$. Similarly, the area $A_2(\lambda)$ of the parallelogram in Fig. 1 (b) is:

$$A_2(\lambda) = \lambda \cdot (-l_x) \cdot (u_x - l_x). \tag{4}$$

$A_2(\lambda)$ is an increasing function of $\lambda$ and also minimized at $\lambda = \frac{u_x}{u_x - l_x}$. In summary, we obtain the following theorem:

**Theorem 3.1** *Let $\mathcal{Z}$ be the input to a ReLU function $y = ReLU(x)$. Consider the set of pointwise Zonotope approximations $\mathcal{O}$ of the output that only alter the affine form $\hat{y}$ of the variable $y$. The new affine form $\hat{y}$ for the output with the minimal area in the $xy$-plane is given by:*

$$\hat{y} = \begin{cases} \hat{x}, & \text{if } l_x > 0, \\ [0, 0], & \text{if } u_x \leq 0, \\ [\lambda_l, \lambda_u] \cdot \hat{x} + [\mu_l, \mu_u] + [\mu_l, \mu_u] \cdot \epsilon_{new}, & \text{otherwise.} \end{cases} \tag{5}$$

Here $\lambda_l, \lambda_u$ are floating point representations of $\lambda_{\text{opt}} = \frac{u_x}{u_x - l_x}$ using rounding towards $-\infty$ and $+\infty$ respectively. Similarly, $\mu_l, \mu_u$ are floating point representations of $\mu = -\frac{u_x \cdot l_x}{2 \cdot (u_x - l_x)}$ using rounding towards $-\infty$ and $+\infty$ respectively, and $\epsilon_{\text{new}} \in [-1, 1]$ is a new noise symbol.

The running time of the optimal transformer in Theorem 3.1 is linear in the number $p$ of noise symbols. One can also define an optimal Zonotope transformer minimizing the volume of the output zonotope, however this is too expensive and the resulting transformer cannot be applied pointwise.

## 3.2 Sigmoid

The effect of applying the Sigmoid function on an input zonotope $\mathcal{Z}$ can be represented with the assignment $y := \sigma(x)$ where $x, y \in \mathcal{X}$ and $\sigma(x) = \frac{e^x}{1 + e^x}$. For the assigned variable $y$, we have $[l_y, u_y] \subseteq [0, 1]$. When $l_x = u_x$, then $\hat{y} := \left[ \frac{e^{u_x}}{1 + e^{u_x}}, \frac{e^{u_x}}{1 + e^{u_x}} \right]$ and the resulting zonotope is exact, otherwise the output cannot be exactly represented by a zonotope and thus approximations are required. We define pointwise approximations for the Sigmoid function such that $l_y = \sigma(l_x), u_y = \sigma(u_x)$ and then choose the one minimizing the area of its projection in the $xy$-plane.

Fig. 2 shows the projections into the $xy$-plane of a set of sound zonotope approximations for the output of the Sigmoid function which have $l_y = \sigma(l_x), u_y = \sigma(u_x)$. As for ReLU, the projections have the shape of a parallelogram with two vertical lines and two parallel lines of slope $\lambda$ which parameterizes the set. To ensure soundness, we have $0 \leq \lambda \leq \min(\sigma'(l_x), \sigma'(u_x))$ where $\sigma'_x = \frac{e^x}{(1 + e^x)^2}$.

The area $A(\lambda)$ of the parallelogram with slope $\lambda$ in Fig. 2 is:

$$A(\lambda) = (\sigma(u_x) - \sigma(l_x) - \lambda \cdot (u_x - l_x)) \cdot (u_x - l_x) \tag{6}$$

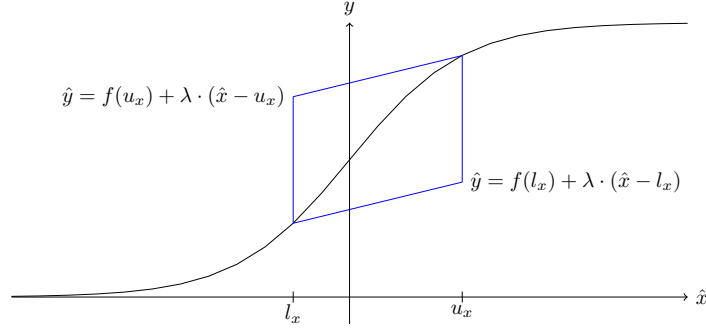

Figure 2: Zonotope approximation for the sigmoid function parameterized by slope $\lambda$, where $0 \leq \lambda \leq \min(f'(l_x), f'(u_x))$.

$A(\lambda)$ is a decreasing function of $\lambda$ and thus $A(\lambda)$ is minimized at $\lambda_{opt} = \min(\sigma'(l_x), \sigma'(u_x))$. This yields the following theorem:

**Theorem 3.2** *Let $\mathcal{Z}$ be the input to a smooth S-shaped[1] function $y = f(x)$ (such as the Sigmoid function $y = \sigma(x) = \frac{e^x}{1+e^x}$). Consider the set of pointwise Zonotope approximations $\mathcal{O}$ of the output that only alter the affine form $\hat{y}$ of the variable $y$ and where the box concretization of $\hat{y}$ satisfies $l_y = \sigma(l_x), u_y = \sigma(u_x)$. The new affine form $\hat{y}$ for the output with the minimum area in the xy-plane is given by:*

$$\hat{y} = \begin{cases} [f(u_x)_l, f(u_x)_u], & \text{if } l_x = u_x, \\ [\lambda_l, \lambda_u] \cdot \hat{x} + [\mu_l^1, \mu_u^1] + [\mu_l^2, \mu_u^2] \cdot \epsilon_{new}, & \text{otherwise,} \end{cases} \quad (7)$$

Here, $f(u_x)_l, f(u_x)_u$ are floating point representations of $f(u_x)$ rounded towards $-\infty$ and $+\infty$ respectively and $\lambda_l, \lambda_u$ are floating point representations of $\lambda_{opt} = \min(f'(l_x), f'(u_x))$ using rounding towards $-\infty$ and $+\infty$ respectively. Similarly $\mu_l^1, \mu_u^1$ and $\mu_l^2, \mu_u^2$ are floating point representations of $\mu_1 = \frac{1}{2}(f(u_x) + f(l_x) - \lambda_{opt} \cdot (u_x + l_x))$ and $\mu_2 = \frac{1}{2}f(u_x) - f(l_x) - \lambda_{opt} \cdot (u_x - l_x)$ computed using rounding towards $-\infty$ and $+\infty$ and adding the error due to the non-associativity of floating point addition, and $\epsilon_{new} \in [-1, 1]$ is a new noise symbol. As with ReLU, the optimal Sigmoid transformer in Theorem 3.2 has linear running time in the number of noise symbols and can be applied pointwise.

### 3.3 Tanh

The Tanh function is also S-shaped, like the Sigmoid function. A fast, optimal, and pointwise Tanh transformer can be defined using Theorem 3.2 by setting $f(x) = \tanh(x)$ and $f'(x) = 1 - \tanh^2(x)$.

## 4 Experiments

We now evaluate the effectiveness of our new Zonotope transformers for verifying local robustness of neural networks. Our implementation is available as an end-to-end automated verifier, called *DeepZ*. The verifier is implemented in Python, however, the underlying abstract transformers are implemented in C (for performance) in both the sequential and the parallel version, and are made available as part of the public ELINA [1, 25] library.

### 4.1 Experimental setup

**Evaluation datasets.** We used the popular MNIST [18] and CIFAR10 [16] datasets for our experiments. MNIST contains $60\,000$ grayscale images of size $28 \times 28$ pixels and CIFAR10 consists of $60\,000$ RGB images of size $32 \times 32$ pixels.

**Neural networks.** Table 1 shows the fully connected feedforward (FFNNs), convolutional (CNNs), and residual networks for the MNIST and CIFAR10 datasets used in our experiments. We used both

Table 1: Neural network architectures used in our experiments.

| Dataset | Model | Type | #Hidden units |
|---|---|---|---|
| MNIST | FFNNSmall | fully connected | 610 |
| | FFNNBig | fully connected | 3 010 |
| | ConvSmall | convolutional | 3 604 |
| | ConvMed | convolutional | 4 804 |
| | ConvBig | convolutional | 34 688 |
| | ConvSuper | convolutional | 88 500 |
| | Skip | residual | 71,650 |
| CIFAR10 | FFNNBig | fully connected | 3 010 |
| | ConvSmall | convolutional | 4 852 |
| | ConvBig | convolutional | 62 464 |

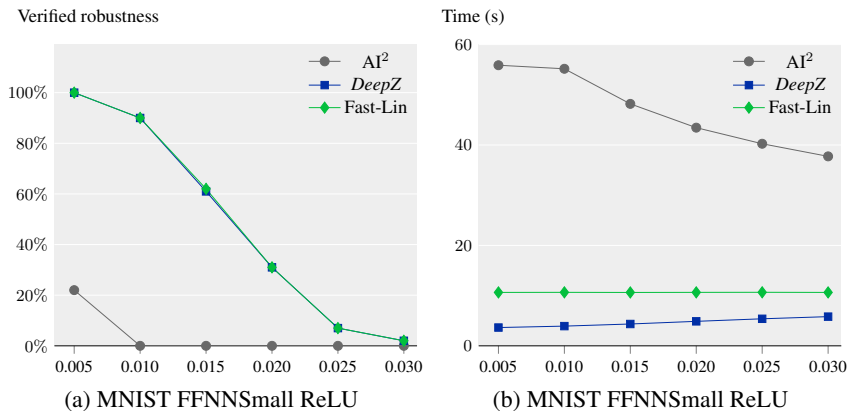

Figure 3: Comparing the performance and precision of *DeepZ* with the state of the art.

undefended and defended training procedures for training our networks. For adversarial training, we used DiffAI [21] and projected gradient descent (PGD) [6] parameterized with $\epsilon$. In our evaluation, we refer to the undefended nets as *Point*, and to the defended networks with the name of the training procedure (either DiffAI or PGD). More details on our neural networks and the training procedures can be found in the appendix.

**Robustness property.** We consider the standard $L_\infty$-norm-based perturbation regions $\mathcal{R}_{\overline{x},\epsilon}$ [3], where $\mathcal{R}_{\overline{x},\epsilon}$ contains all perturbed inputs $\overline{x}'$ where each pixel $\overline{x}'_i$ has a distance of at most $\epsilon$ from the corresponding pixel $\overline{x}_i$ in the original input $\overline{x}$. $\mathcal{R}_{\overline{x},\epsilon}$ can be exactly represented by a single zonotope.

**Benchmarks.** We selected the first 100 images from the test set of each data set. Then, we specified a robustness property for each image using a set of robustness bounds $\epsilon$.

## 4.2 Experimental results

All experiments for the FFNNs were carried out on a 3.3 GHz 10 core Intel i9-7900X Skylake CPU with 64 GB main memory; the CNNs and the residual network were evaluated on a 2.6 GHz 14 core Intel Xeon CPU E5-2690 with 512 GB main memory. We used a time limit of 10 minutes per run for all our experiments.

**Comparison with prior work.** We compare the precision and performance of the sequential version of *DeepZ* against two state-of-the-art certifiers Fast-Lin [29] and AI[2] [10] on the FFNNSmall MNIST network with ReLU activation. We note that both of these certifiers support only a subset of the network architectures that *DeepZ* can support. Specifically, Fast-Lin only supports FFNNs with ReLU activations whereas AI[2] supports FFNNs and CNNs with ReLU activations. We also note that Fast-Lin is not sound under floating point semantics.

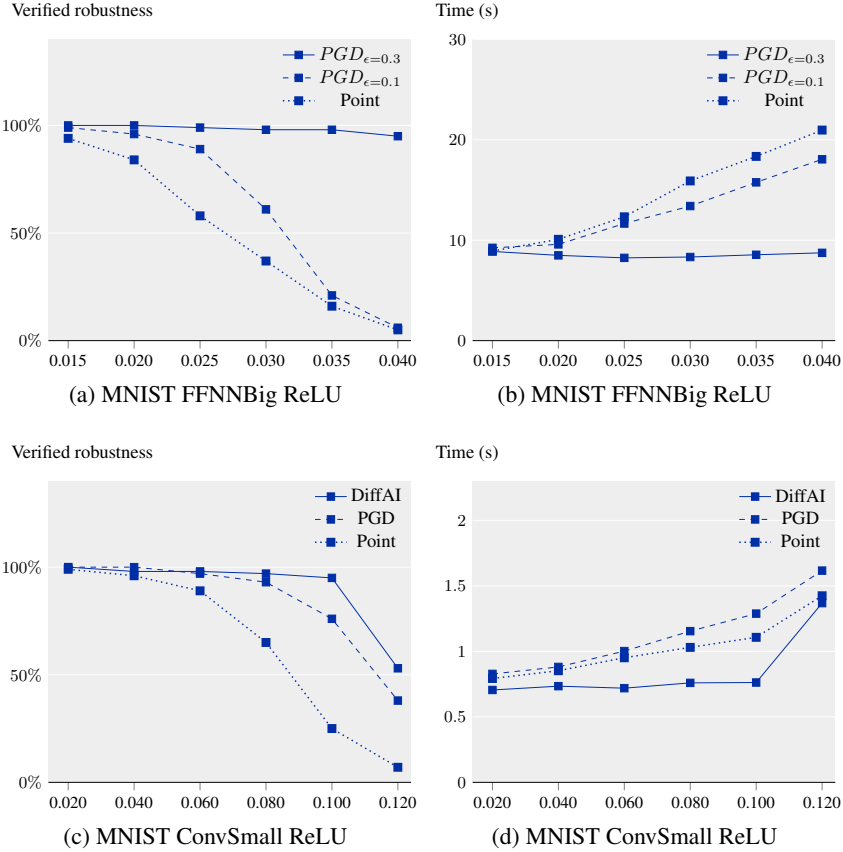

Figure 4: Verified robustness by *DeepZ* on the MNIST networks with ReLU activations.

Table 2: Verified robustness by *DeepZ* on the large networks trained with DiffAI.

| Dataset | Model | $\epsilon$ | % verified robustness | average runtime(s) |
|---------|-------|------------|----------------------|--------------------|
| MNIST | ConvBig | 0.1 | 97 | 5 |
| | ConvBig | 0.2 | 79 | 7 |
| | ConvBig | 0.3 | 37 | 17 |
| | ConvSuper | 0.1 | 97 | 133 |
| | Skip | 0.1 | 95 | 29 |
| CIFAR10 | ConvBig | 0.006 | 50 | 39 |

Fig. 3 shows the percentage of verified robustness and the average analysis time of all three certifiers. The values of $\epsilon$ are shown on the x-axis. *DeepZ* has the same precision as Fast-Lin but is up to 2.5x times faster. We note that the runtime of *DeepZ* increases with increasing value of $\epsilon$; this is because the complexity of our analysis is determined by the maximum number of noise symbols in the affine form. Our ReLU transformer creates one noise symbol for any variable that can take both positive and negative values. The number of such cases rises with the increasing value of $\epsilon$. On the other hand, AI[2] is significantly less precise and slower compared to both Fast-Lin and *DeepZ*. We also compared *DeepZ* against the duality-based certifier from [7], however it always timed out in our experiments.

**Detailed experiments.** Next, we evaluate *DeepZ* on the remaining networks using the parallelized version of our Zonotope transformers. Fig. 4 shows the percentage of verified robustness and the average analysis time of *DeepZ* for the MNIST networks with ReLU activations. *DeepZ* analyzes all FFNNBig networks with average runtime $\leq 22$ seconds and proves 95% of the robustness properties for $\epsilon = 0.04$ for the defended $PGD_{\epsilon=0.3}$ network. *DeepZ* is able to analyze all ConvSmall networks with average runtime $\leq 2$ seconds. It proves 95% of the robustness properties for $\epsilon = 0.1$ on the

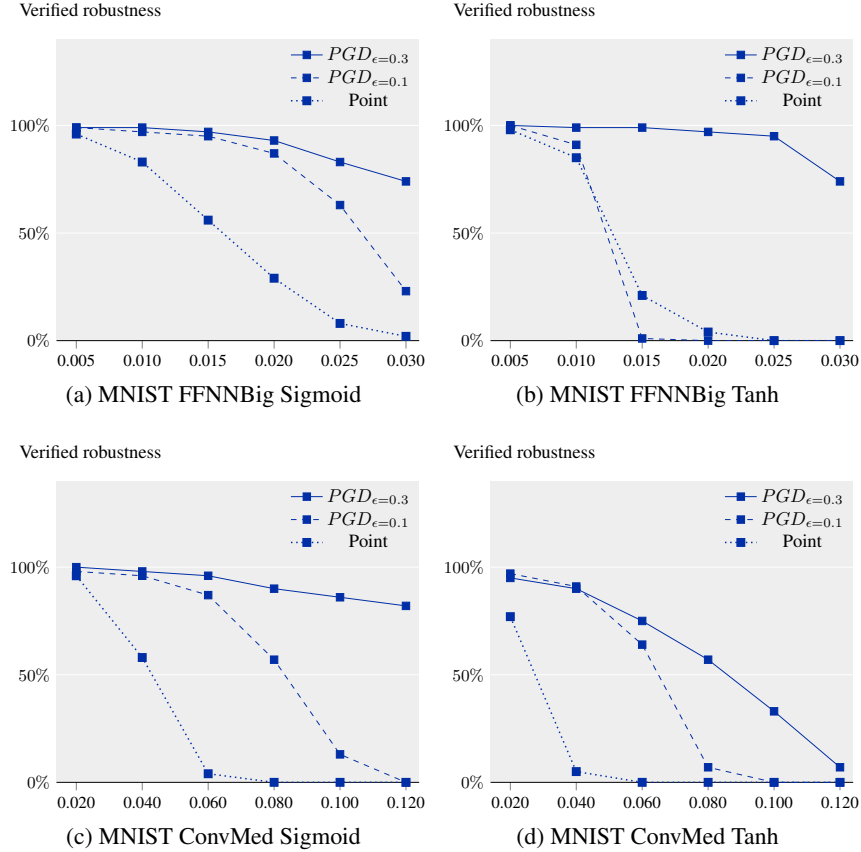

Figure 5: Verified robustness by *DeepZ* on the MNIST networks with Sigmoid and Tanh activations.

ConvSmall network defended with DiffAI. Table 2 shows the precision and the performance of *DeepZ* on ConvBig, ConvSuper, and Skip networks trained with DiffAI. *DeepZ* proves 95% and 97% of robustness properties for the Skip and ConvSuper networks containing $> 71,000$ and $> 88\,000$ hidden units respectively in 29 and 133 seconds on average.

Fig. 5 shows the precision and the performance of *DeepZ* on the MNIST FFNNBig and ConvMed networks with Sigmoid and Tanh activations. It can be seen that *DeepZ* verifies 74% of the robustness properties on the FFNNBig Sigmoid and Tanh networks trained with $PGD_{\epsilon=0.3}$ for $\epsilon = 0.03$. *DeepZ* verifies 82% of the robustness properties on the ConvMed Sigmoid network for $\epsilon = 0.1$. The corresponding number for the Tanh network is 33%. We note that unlike the ReLU transformer, both Sigmoid and Tanh transformers always create a new noise symbol whenever $l_x \neq u_x$. Thus, the runtime does not increase significantly with $\epsilon$ and is not plotted. *DeepZ* has an average runtime of $\leq 35$ and $\leq 22$ seconds on all FFNNBig and ConvMed networks, respectively.

Fig. 6 shows that *DeepZ* has an average runtime of $\leq 50$ seconds for the CIFAR10 FFNNBig ReLU networks. It can be seen that the defended FFNNBig CIFAR10 ReLU networks are not significantly more provable than the undefended network. However, *DeepZ* verifies more properties on the defended ConvSmall networks than the undefended one and proves 75% of robustness properties on the DiffAI defended network for $\epsilon = 0.01$. *DeepZ* has an average runtime of $\leq 3$ seconds on all ConvSmall networks. *DeepZ* is able to verify 50% of robustness properties for ConvBig network defended with DiffAI with an average runtime of 39 seconds as shown in Table 2.

*DeepZ* verifies 82% of robustness properties on the FFNNBig Sigmoid network defended with $PGD_{\epsilon=0.0078}$ for $\epsilon = 0.012$ in Fig. 7. It verifies 46% of the robustness properties on the FFNNBig network with Tanh activation trained using $PGD_{\epsilon=0.0313}$ for the same $\epsilon$. The average runtime of *DeepZ* on all networks is $\leq 90$ seconds.

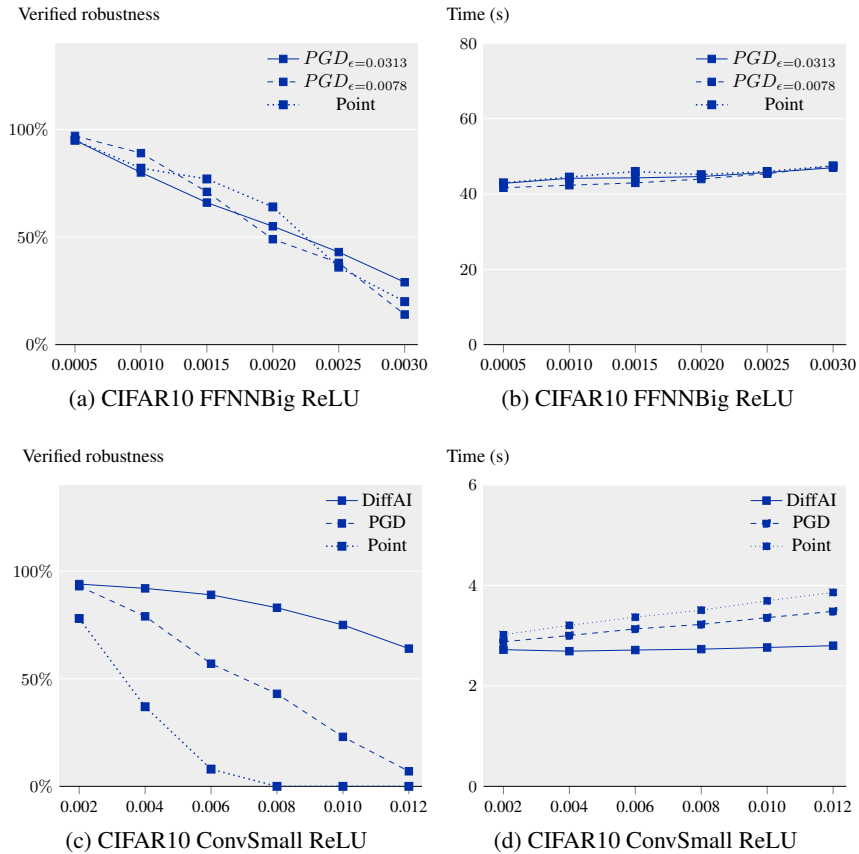

Figure 6: Verified robustness by *DeepZ* on the CIFAR10 networks with ReLU activations.

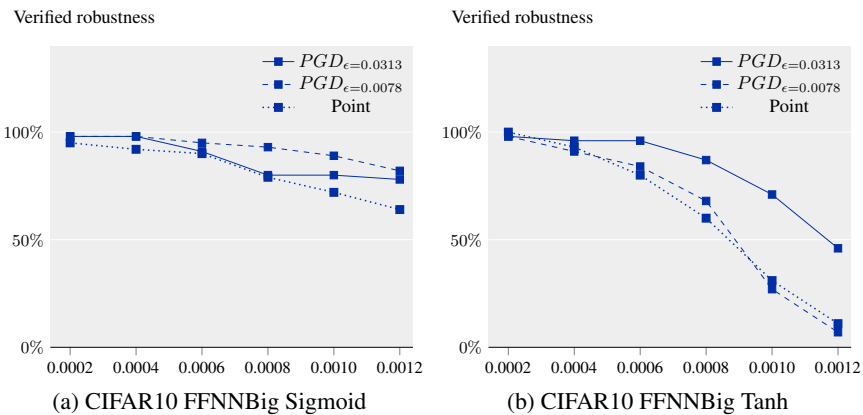

Figure 7: Verified robustness by *DeepZ* on the CIFAR10 networks with Sigmoid and Tanh activations.

# 5 Conclusion

We introduced fast and precise Zonotope abstract transformers for key non-linear activations used in modern neural networks. We used these transformers to build *DeepZ*, an automated verifier for proving the robustness of neural networks against adversarial attacks. We evaluated the effectiveness of *DeepZ* on verifying robustness of large feedforward, convolutional, and residual networks against challenging $L_\infty$-norm attacks. Our results show that *DeepZ* is more precise and faster than prior work, while also ensuring soundness with respect to floating point operations.

## Footnotes

[1]A smooth function $f \colon \mathbb{R} \to \mathbb{R}$ is said to be S-shaped if $f'(x) \geq 0$ and there exists a value $x'$ such that for all $x \in \mathbb{R}$, we have $f''(x) \leq 0 \Leftrightarrow x \leq x'$.

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

# A  Dataset Normalization

For each dataset we include a normalization layer (which gets applied *after* the $\epsilon$-sized box has been calculated) using an approximated mean $\mu$ and standard deviation $\sigma$ per channel as $\frac{X-\mu}{\sigma}$.

**MNIST:** $\mu = 0.1307,\ \sigma = 0.3081$.

**CIFAR10:** $\mu = [0.4914, 0.4822, 0.4465],\ \sigma = [0.2023, 0.1994, 0.2010]$.

# B  Neural Networks Evaluated

We test with six networks: one feed forward, four convolutional (without maxpool), and one with a residual connection. These are trained in various ways. In the following descriptions, we use $\text{Conv}_s C \times W \times H$ to mean a convolutional layer that outputs $C$ channels, with a kernel width of $W$ pixels and height of $H$, with a stride of $s$ which then applies ReLU to every output. FC $n$ is a fully connected layer which outputs $n$ neurons without automatically applying ReLU.

For each architecture we test three versions: (i) an undefended network; (ii) a network defended with MI-FGSM (a PGD variant which we refer to as PGD in the graphs) [6] with $\mu = 1$, 22 iterations and two restarts, where the step size is $\epsilon = 5.5^{-1}$ for the $\epsilon$ used for training; (iii) a network defended with a system based on DiffAI [21].

**FFNN.** A 6 layer feed forward net with 500 nodes in each and an activation (ReLU, Sigmoid, Tanh) after each layer except the last.

**ConvSmall.** Our smallest convolutional network with no convolutional padding.

$$x \to \text{Conv}_2 16 \times 4 \times 4 \to \text{ReLU} \to \text{Conv}_2 32 \times 4 \times 4 \to \text{ReLU} \to \text{FC } 100 \to z.$$

**ConvMed.** Similar to ConvSmall, but with a convolutional padding of 1. Here we test with the three activations Act = ReLU, Sigmoid, and Tanh.

$$x \to \text{Conv}_2 16 \times 4 \times 4 \to \text{Act} \to \text{Conv}_2 32 \times 4 \times 4 \to \text{Act} \to \text{FC } 1000 \to z.$$

**ConvBig.** A significantly larger convolutional network with a convolutional padding of 1.

$$\begin{aligned}
x &\to \text{Conv}_1 32 \times 3 \times 3 \to \text{ReLU} \to \text{Conv}_2 32 \times 4 \times 4 \to \text{ReLU} \\
&\to \text{Conv}_1 64 \times 3 \times 3 \to \text{ReLU} \to \text{Conv}_2 64 \times 4 \times 4 \to \text{ReLU} \\
&\to \text{FC } 512 \to \text{ReLU} \to \text{FC } 512 \to z.
\end{aligned}$$

**ConvSuper**  Our largest convolutional network with no padding.

$$\begin{aligned}
x &\to \text{Conv}_1 32 \times 3 \times 3 \to \text{Conv}_1 32 \times 4 \times 4 \\
&\to \text{Conv}_1 64 \times 3 \times 3 \to \text{Conv}_1 64 \times 4 \times 4 \\
&\to \text{FC } 512 \to \text{ReLU} \to \text{FC } 512 \to z.
\end{aligned}$$

**Skip**  Two convolutional networks of different sizes, which are then concatenated together. This network uses no convolutional padding.

$$\begin{aligned}
x &\to \text{Conv}_1 16 \times 3 \times 3 \to \text{ReLU} \\
&\to \text{Conv}_1 16 \times 3 \times 3 \to \text{ReLU} \\
&\to \text{Conv}_1 32 \times 3 \times 3 \to \text{ReLU} \to \text{FC } 200 \to o_1, \\
x &\to \text{Conv}_1 32 \times 4 \times 4 \to \text{ReLU} \\
&\to \text{Conv}_1 32 \times 4 \times 4 \to \text{ReLU} \to \text{FC } 200 \to o_2, \\
\text{CAT}(o_1, o_2) &\to \text{ReLU} \to \text{FC } 200 \to \text{ReLU} \qquad\quad \to z.
\end{aligned}$$

