[Reviews · NeurIPS 2018]

Reviewer 1



The contribution approximates activation of NNs by zonotopes, ie affine combinations of intervals of possible input-output relationship. Unlike previous work, the computation is efficient. It enables a test towards robustness of a deep network, thus addressing adversarial examples. The idea is to compute the effect of input perturbations based on this abstraction directly. The crucial question is then how to approximate nonlinearities and how to do so efficiently. Results show that the proposed method is more efficient and effective than a previous one. Yet, it is not so clear which adversarial attacks are prevented this way, a bit more comments would be welcome. Also, the work is rather straightforward. The article seems pretty ok as regards the question to robust computation, since it builds on the classical work of zonotopes and it formally allows to identify robustness by means of formal interval arithmetics. The proposed novel approximation of the common functions rely, sigmoid, maxpooling seem straightforward yet doing their work. The auhtors specify why this is novel in clear terms in the rebuttal - so the work contributes a possibly useful extension. to previous work My main critic concerns the claim that adversarial attacks are addressed this way. The authors reply that this is a robustification against standard attacks (basically there does exist a small region around the given data where the network provably yields the same results). How will this affect attack defense in practice? A partial answer is given if (as the authors reply) robustly trained networks are used. Yet, I would like to see the bevaior against adversarial attacks - as an example, I would expect experimental work which demonstrates the distribution of adversarial attacks with respect to the region of robustness which is verified by the model, or checks which demonstrate that such a verified network is more robust in a quantitative way to the existing attack mechanisms. So I think the work is interesting as regards robustification, but I would expect a demo how this can be lined back to robustness to adversarial attacks in practice for the NIPS community

Reviewer 2



The paper develops a new approach for the problem of verifying input-output properties of neural network, an emerging area that has received significant interest and attention in the ML community recently. The approach builds on previous work that uses abstract interpretation (sound abstractions of the transformations at each layer of a neural network) that can be used to reason about worst-case behavior of the network with respect to desired input-output properties (for example, that the network should continue to produce the same label despite adversarially chosen small perturbations of the input). If the property is satisfied by the sound abstraction, the property will be satisfied by the underlying network as well. Strengths of the paper: Well written - the central idea of zonotopes is clearly explained, and the property of area minimizing parallelogram abstractions of various neural network transformations (relu, sigmoid, tanh) is explained geometrically and derived. The paper computes certificates that are sound even accounting for floating point arithmetic - a property that many competing approaches (like those of Kolter et al, Raghunathan et al, Dvijotham et al) do not satisfy. The paper applies to a broad class of transformations commonly found in feedforward neural networks, unlike previous approaches from that use tools from the formal methods community (like Katz et al, Elhers et al etc.) Weaknesses of the paper: Justifying the use of zonotopes: The authors claim that zonotopes provide a sound abstraction that offers the right tradeoff between precision and computation, but do not provide much justification for why this is the case. For example, it was not clear to me why enforcing a parallelogram shape in the abstraction is necessary - for example, the ReLU is abstract better by the triangle used in the PLANET (Elhers et al) paper. I understand that this helps preserve the affine interval expression structure - however, it is not clear why this interval expression structure is crucial for the approach to work or scale. Comparison to prior work: When specialized to the case of piecewise linear activation functions, the approach seems to be equivalent (or very close to) that of https://www.icml.cc/Conferences/2018/Schedule?showEvent=2976 . This paper also proves that the approach is equivalent to a that form (Kolter and Wang, 2017) https://arxiv.org/abs/1711.00851 . While the authors extend the approach to nonlinear activation functions and ensure that the computations are rigorous under finite precision arithmetic, a careful discussion and comparison to this previous work is certainly warranted. Without it, the novelty of the approach in the paper comes under question. Experimental evaluation: I have several concerns with the experimental evaluation section of the paper: It is well-understood from the works of (Kolter and Wang, 2017) and (Raghunathan et al, 2017) that networks that are trained to be verifiable are significantly easier to verify than networks trained arbitrarily. The Given how close the approach is to that from https://www.icml.cc/Conferences/2018/Schedule?showEvent=2976, a comparison to this approach on ReLU networks is certainly warranted. It would be good to compare against other approaches that use formal methods and are sound to floating point arithmetic (like ReLUplex from Katz et al, PLANET from Elhers et al). In the comparison against [Dvijotham et al, 2018], the authors only implement the version of the algorithm with optimization based bound propagation. The approach also works with interval arithmetic based bound propagation and is likely much faster in this variant - Final results: Published numbers for verified robustness against adversarial attacks on MNIST and CIFAR (from https://www.icml.cc/Conferences/2018/Schedule?showEvent=2976 and Kolter and Wang) are significantly better than those from the paper - for example, these papers establish verified error rates of around 5% against perturbations of size .1 or larger while those in the paper are significiantly weaker. While this is likely due to the training procedure used, it would have been good for the authors to apply their methods to the networks trained by (Kolter and Wang), (Mardry et al) which are publicly available to see if they indeed produce state of the art numbers on verified accuracy/robustness. Another possibility is the need to account soundly for finite precision arithmetic, but the reviewer feels that this needs to be justified through numerical studies (or by implementing a version of the authors’ approach that does not guarantee soundness against finite precision, only against precise real arithmetic). Other specifications on the input: The examples considered in the paper assume that the constraints on the input can be given by lower and upper bounds. It is clear that the zonotopes can work with other types of input constraints - however, it is not clear how gracefully the performance (in terms of precision loss) would degrade (for example, with an l2 constraint on the inputs instead of linf). As a side note, at least in the ML community, “zonotopes” is not a commonly used term. Given the central idea of the paper can be captured in the idea of bounding nonlinear tranforms with affine interval expressions, perhaps a simpler name is more appropriate for the ML community? Overall evaluation: Clarity: The paper is clearly written. Quality: The theoretical results are accurate and well presented. However, the quality of the experimental section can be significantly improved. Novelty: The paper adds novelty in addressing finite precision arithmetic and general activation functions relative to previous verification approaches. However, the lack of a clear comparison to https://www.icml.cc/Conferences/2018/Schedule?showEvent=2976 casts a shadow on how novel the core idea of zonotopes, particularly in the context of neural network verification. Significance: The paper is an interesting albeit marginal addition to the state of the art, in that the paper offers a scalable/general approach to robustness certification that is sound against floating point arithmetic.

Reviewer 3



This paper provides an efficient method for certifying the robustness of a neural network to various forms of perturbations. This approach is much more efficient than AI^2 (the prior state of the art, to the best of my knowledge) on standard fully-connected neural networks, and is effective even with convolutions when AI^2 is not. This approach is effective on neural networks with tens of thousands of neurons, a significant improvement on prior work. This paper further extends prior work by allowing both sigmoid and tanh units. While these are not extensively used in neural networks, it is still a worthwhile contribution to provide this analysis. Given this efficiency, I wonder why the authors only evaluate on 10 different starting input images. This seems like a good place to try and evaluate on many more (100? maybe more?) samples to produces more significant results. Similarly, for some figures (e.g., Figure 3) it would be useful to continue the plot further to the right. At what point does the robustness begin to decrease? eps=0.085 is very small on MNIST, when typically the adversarial example literature uses eps=0.3. It does not look like runtime increases substantially as eps is increased, it would be interesting to see if this holds true for larger budgets. Figure 4(c) would be especially interesting. Does the 400_200_100_100 network continue to take longer to run when the verification bound increases? Given that it reaches 0% accuracy at eps=0.024, does the runtime increase for larger epsilon values? This paper could be further improved by giving upper bounds on robustness with PGD (or other optimization-based attacks). If the certifications in this paper are lower bounds, an upper bound through an optimization based attack would provide evidence that the lower bounds were precise --- in particular, it may even be the case that it is 100% precise in some cases (where the lower bound and upper bounds might give the same results). For example, that analysis would help discover if the tanh networks are the least robust, or if it is just that they are the hardest to prove robustness bounds for. (The 400-200-100-100 network, for example, can give 0% certified robustness at eps=.008 -- is this because the robustness is lower or just that it is harder to prove properties on).